# Synergistic Attraction and Ecological Effects of Multi-Source Physical and Chemical Trapping Methods with Different Mechanism Combinations on Rice Pests

**DOI:** 10.3390/insects16101001

**Published:** 2025-09-26

**Authors:** Wei Zeng, Jianping Peng, Chuanhong Feng, Qinghua Chen, Chunxian Jiang

**Affiliations:** 1Plant Protection Station of Dachuan District, Dazhou 635711, China; 2Plant Protection Station of Agricultural and Rural Department of Sichuan Province, Chengdu 610041, China; fengchuanhong8@163.com; 3Key Laboratory of Integrated Pest Management on Crops in Southwest, Ministry of Agriculture, Institute of Plant Protection, Sichuan Academy of Agricultural Sciences, Chengdu 610066, China; 13980767793@139.com; 4College of Agronomy, Sichuan Agricultural University, Chengdu 611130, China; chunxianjiang@126.com

**Keywords:** color-sex-light trap, multi-source physical and chemical combination, attraction measurement device, rice pests, attracting measurement effect

## Abstract

The use of accurate and efficient monitoring methods to detect and grasp insect situations early is the basis and prerequisite for scientific and effective integrated pest management (IPM). For the main rice pests, we used a multi-source physical and chemical trapping method based on different mechanism combinations such as color plates, insect sex pheromones, transmitting (shielding) light covers, and solar-powered automatic insect-attracting ultraviolet lamps. This study evaluated the synergistic attracting effects and ecological impacts of different methods on major rice pests, as well as screened and provided monitoring methods suitable for major rice pests, and the positive synergistic effects and negative effects of attracting methods based on different mechanism combinations on pest attraction, and then adopted accurate and efficient pest monitoring methods and strategies in production, which is conducive to improving the effectiveness of integrated crop pest management.

## 1. Introduction

Crop pests and diseases have always been a major obstacle to agricultural production, restricting agricultural production efficiency and quality. At the same time, their chemical control also threatens the ecological environment [1]. According to the Food and Agriculture Organization (FAO) of the United Nations, the annual global food production loss due to pests and diseases accounts for 30% of the global food production [2,3,4,5]. Rice is the main component of the daily diet of nearly half of the world’s population [6,7] and is the world’s third largest grain after corn and wheat. In China, there are more than 90 common pests and diseases that harm rice [8]. Continuously exploring convenient, reliable, simplified, practical, or intelligent rice insect monitoring methods, technologies, and equipment will further improve the efficiency of accurate monitoring and terminal detection capabilities, which is of great significance for the scientific and timely guidance of pest control and for reducing disaster losses.

Internationally, conventional monitoring methods for adult rice pests include the light trap method, insect sex pheromone trap method, field moth-chasing method, and plant-flapping method. Different methods have their own advantages and characteristics [9,10]. The light trap method does not respond strongly to some weakly phototactic pests [11,12]. The insect sex pheromone trap method has the advantages of specificity and sensitivity [12,13,14], but for pests such as *C. suppressalis*, there are differences in the effects of trapping different generations. For example, this method is most effective or sensitive in trapping the overwintering generations of adults of the *C. suppressalis*, but the number of adults of the second and third generations trapped is lower than that of light trapping [12,15]. Field monitoring methods such as the manual field moth-chasing method and the plant-flapping method are labor-intensive and time-consuming [12,16,17]. In recent years, intelligent automated monitoring and reporting equipment based on light trapping or sex trapping has greatly improved the accuracy and timeliness of monitoring and early warning [18,19]. However, further optimization is needed in terms of automatic counting accuracy and recognition accuracy of automatically collected data [19,20].

With people paying more and more attention to food safety, the viscose color plate trapping method has attracted widespread attention [21]. It is a technology that uses the insect’s attraction to a certain color to trap and kill pests. It has the advantages of being environmentally friendly (reducing the amount of pesticide used) and simple [16]. It has been well studied and applied in trapping and controlling target pests of various crops [16,22]. In addition, the combination of the viscose color plate trapping method and the sex trapping method has the characteristics of clear insect bodies, no overlapping interference, and easy identification of pests such as *P. auttata* and *C. suppressalis* [23,24,25]. The composite integration and utilization of the characteristics and complementary advantages of different trapping methods are conducive to the precise monitoring and control of various rice pests.

To ensure accurate and convenient monitoring of rice insect infestations and improve pest monitoring efficiency during rice cultivation, this research team designed a multi-source physical and chemical trapping method and integrated a device based on a combination of different mechanisms, including color plates, insect sex pheromones, transmitting (shielding) light covers, and solar-powered automatic insect-attracting ultraviolet lamps. The goal was to clarify the synergistic attracting effects and ecological impacts of different multi-source physical and chemical trapping methods on adult rice pests, providing a scientific basis for the rational and effective application of multi-source physical and chemical trapping for the monitoring and pest management of major crop pests.

## 2. Materials and Methods

### 2.1. Test Tools

Experimental devices for different methods were designed and manufactured based on an innovative pest monitoring method and device developed by the research team [26,27]. The device is mainly composed of a color plate, an insect sex pheromone, a rainproof and dustproof cover, and its supporting materials. The insect sticky color plate was designed with insect sticky plate perforations on which different insect sex pheromone core tubes can be directly and flexibly installed and fixed. Automatic solar-powered insect-attracting ultraviolet lamps (ul, purple light, 0.36 W, 365 nm, produced by Zhongjie Sifang (Beijing) Biotechnology Co., Ltd., Beijing, China), with light control, rain control, and other sensors, and an automatic adjustment system were fixedly installed above the rain and dustproof cover of the color plate + insect sex pheromone trap core combination device. This constitutes a multi-source physical and chemical trapping method and integrated device composed of a light green sticky insect plate (cp) + insect pheromone lure (sp) + rain and dustproof light shield (transparent) cover (sc or lc) + solar-powered automatic insect-attracting ultraviolet lamps (ul), which is a color plate, sex trap, and light trap device (triple-trap device or cp + sp + s(l)c + ul). In this device, if the sticky insect color plate and the insect sex information attractant core are installed, the device is simply called a color attractant and sex attractant device (double-trap device or cp + sp + s(l)c). If only the insect sticking color plate is installed, the device is simply called a light green sticky plate device or cp + s(l)c. According to the two different materials used for the rain and dust cover in each device, colorless transparent or opaque, each device is designed as a light-transmitting type (lc) and a light-shielding type (sc). Different types of pest trapping devices are shown in Figure 1**,** which were produced or processed by Sichuan Wuhua Agricultural Technology Co., Ltd. (Chengdu, China).

The sticky insect color plates used in the experiment were all light green sticky plate with a size of 20 cm × 25 cm, a visible light wavelength of about 550 nm, and a thickness of 0.3 mm ± 0.02 mm. The boat-shaped glue trap (st) used in the experiment was equipped with a sex pheromone lure for *C. suppressalis* (sp(Cs)) or a sex pheromone lure for *C. medinalis* (sp(Cm)), referred to as a glue-type sex lure device (st-sp(Cs) or st-sp(Cm)). The sex pheromone lures for *C. suppressalis* and *C. medinalis* used were all PVC capillaries. Among them, the main components of the sex pheromone lure for *C. medinalis* were (Z)-13-octadecenal, (Z)-11-octadecenal, (Z)-13-octadecen-1-ol, and (Z)-11-octadecen-1-ol, with a total content of 740 μg/piece and a lasting effect of ≥60 days. The main ingredients of the *C. suppressalis* sex pheromone lure core were 0.5% (Z)-11-hexadecenal and 0.05% (Z)-9-hexadecenal, with a total effective ingredient content of 0.55% and a lasting effect of ≥60 days. The glue-type sex lure device trap and lure were produced by Zhejiang Ningbo Niukang Biotechnology Co., Ltd. (Ningbo, China).

### 2.2. Test Location and Basic Information

Test site 1 was a rice planting area of Wange Community 10, Heshi Town, Dachuan District, Dazhou City, Sichuan Province (31°10′18″ N, 107°25′46″ E). It is a river valley and shallow hill area. The altitude is 302.0–310.0 m. Rice was sown in mid-to-late March, transplanted in early to mid-May, and harvested in late August. This region is home to common pests such as *C. suppressalis*, migratory *S. furcifera*, and *C. medinalis. C. suppressalis* is the primary pest in the early growth phase of rice, while migratory *C. suppressalis* and *C. medinalis* dominate the mid-to-late growth phase. The incidence of migratory pests fluctuates significantly from year to year, depending on the timing and number of migratory pests. Corn, vegetables, citrus, pears, plums, and other grain and cash crops are cultivated near the rice-growing areas.

Test site 2 was a rice planting area of Liujiaba Village 1, Maliu Town, Dachuan District, Dazhou City, Sichuan Province (31°1′39″ N, 107°40′34″ E). It is a rice field with rice (wheat and cole) rotation in Pingba area, about 343.6 m above sea level. In mid-to-late May, rice seedlings were transplanted or rice was directly sown after the early spring crops were harvested, and harvested in mid-to-late September. This rice-growing area is home to migratory pests (*C. medinalis* and *S. furcifera*) and common pests such as *C. suppressalis*, *P. auttata*, and rice leafhopper. In the middle and late stages of the rice season, the main pests were *C. suppressalis*, rice leafhopper, and two migratory pests.

### 2.3. Experimental Design and Methods

During the experimental period, three representative, non-adjacent rice fields were selected at each test site as replicate plots. One set of the corresponding experimental devices was randomly installed in each field. Each device was installed 50 cm from the edge of the rice field, with a spacing of at least 8 m between devices. Based on the occurrence patterns and characteristics of major insect pests such as *C. medinalis* and *C. suppressalis* in each test site, the corresponding insect sex pheromones were installed on the relevant devices. During the rice seedling stage, the bottom of the device color plate was maintained at a height of 50 cm above the field surface. During the vegetative growth period of rice, when installing the sp(Cm), the bottom of the color plate should be about 10 cm away from the canopy leaf surface, and when installing the sp(Cs), the bottom of the color plate should be about 25 cm away from the canopy leaf surface. Surveys were conducted approximately every seven days. During the field trial, researchers identified and classified crop pests, natural enemies, beneficial insects, and non-target pests based on their typical morphological characteristics and counted their numbers. Sticky color plates were promptly replaced after the surveys. Insect sex pheromones were not replaced during the experimental period until the end of the surveys. No pesticides were applied, and other management measures remained largely unchanged.

The specific experimental design of the simultaneous comparison test of different types of trapping devices is shown in Table 1.

#### 2.3.1. Comparison of Light-Shielding Triple-Trap Device and Double-Trap Devices for Trapping Migratory Pests

Field trial 1: 13 July–21 August 2023, at test site 1. The fifth (third) generation of adults of the main pest, *S. furcifera*, were observed at a large scale at the test site [11,28]. Both the light-shielding triple-trap device (cp + sp(Cm) + sc + ul) and light-shielding double-trap device (cp + sp(Cm) + sc) underwent a synchronous comparative trap test. The trapping effects of the two devices on two migratory rice pests (*C. medinalis* and *S. furcifera*) were compared.

Field trial 2: 1–23 August 2023, at test site 2. The fifth (third) generation of the main pest of rice leaf roller occurred seriously [11,29]. Using the commonly used glue-type sex lure device (st-sp(Cm)) in production as a control, the devices of field trial 1 were also used to carry out the synchronous comparative attractant test to investigate the luring effect of the three devices on major mid- and late-stage pests such as *C. medinalis*, beneficial insects (natural enemies), and major non-target *Muscomorpha*.

#### 2.3.2. Three Types Light-Transmitting Trapping Devices Were Used to Compare and Simultaneously Trap Major Pests

From April to August 2024, at test site 1, using the commonly used glue-type sex lure device (st-sp) in production as a control, the light-transmitting triple-trap device (cp + sp + lc + ul), light-transmitting double-trap device (cp + sp + lc), and light-transmitting light green sticky plate device (cp + lc) were selected to undergo synchronous comparative attractant tests. Two field trials were conducted.

Field trial 3: 13 April–9 June 2024. During the period when the overwintering generation of adult *C.suppressalis*, sp(Cs) were used to compare and investigate the effects of four devices (cp + sp(Cs) + lc + ul, cp + sp(Cs) + lc, cp + lc, st-sp(Cs)) on trapping major pests such as *C.suppressalis*, other crop pests, beneficial insects such as natural enemies, and major non-target *Muscomorpha*.

Field trial 4: 9 June–19 August 2024. sp(Cs) was replaced with sp(Cm), and the trapping was continued to be compared synchronously to explore the trapping effects of the four devices (cp + sp(Cm) + lc + ul, cp + sp(Cm) + lc, cp + lc, st-sp(Cm)) on migratory pests and other major pests of rice in the middle and late stages.

#### 2.3.3. Comparison of Trapping Effects of Light-Shielding and Light-Transmitting Triple-Trap Devices for Main Pests in Middle and Late Stages of Rice Production

From July to September 2024, at test site 2, two devices, the light-shielding triple-trap device (cp + sp + sc + ul) and light-transmitting triple-trap device (cp + sp + lc + ul), were used to conduct the simultaneous comparative trapping of major rice pests in the middle and late stages. Based on the occurrence and characteristics of *C. medinalis* and *C. suppressalis* in the field, two tests were carried out successively.

Field trial 5: 28 July–21 August 2024. sp(Cm) were used for trapping. The effects of two devices (cp + sp(Cm) + sc + ul, cp + sp(Cm) + lc + ul) on trapping major rice pests, natural enemies, and other beneficial insects, as well as major non-target *Muscomorpha*, were investigated in the middle and late stages of rice production.

Field trial 6: 21 August–6 September 2024. sp(Cs) were used for trapping. As in field trial 5, the trapping effectiveness of the two devices (cp + sp(Cs) + sc + ul, cp + sp(Cs) + lc + ul) was compared.

### 2.4. Data Analysis

The data processing system (DPS) and software version V18.10 [30] were used to conduct a *t*-test to test the significance of the difference between the means of the two groups of experimental sample data. The number of major pests of rice and other crops, natural enemies, and major non-target flies trapped by three or more different experimental devices was subjected to a randomized block single-factor analysis of variance or a significant difference analysis. In the variance analysis, the data were not transformed, and Duncan’s new multiple range method was used for multiple comparisons.

## 3. Results

### 3.1. Comparison of the Detection Effects of the Light-Shielding-Type Triple-Trap Device and Double-Trap Device

In field trial 1, the total number of rice pests, *S. furcifera* and *C. medinalis*, trapped by the triple-trap device (64.33 ± 3.53) was higher than that trapped with the double-trap device (33.33 ± 10.59) (*p* = 0.0499) (Figure 2A). For the target pest, *C. medinalis*, the triple-trap device captured fewer insects (2.33 ± 0.33) than the double-trap device (7.00 ± 0.58) (*p* = 0.0022). For *S. furcifera*, the triple-trap device captured more insects (62.00 ± 3.79) than the double-trap device (26.33 ± 10.73) (*p* = 0.0350) (Figure 2B). The two trapping devices showed consistent results in detecting the dynamic trends of the fifth (third) generation of *S. furcifera* adult populations and the peak period of adulthood (r = 0.8523) (Figure 2C).

### 3.2. Comparison of Detection Effects of the Light-Shielding Triple-Trap Device, the Double-Trap Device, and the Glue-Type Sex Lure Device

In field trial 2, the relative abundance of rice pests trapped in all devices was highest for *C. medinalis*. In addition, the relative abundance of *C. medinalis* was higher in the glue-type sex lure device than in the other two devices (the triple-trap device and double-trap device). The relative abundance of *C. suppressalis*, *P. auttata*, and rice leafhoppers was higher in the triple-trap device than in the other two devices (the double-trap device and glue-type sex lure device) (Figure 3A). In terms of the number of rice pest species trapped, the triple-trap device captured two and three more species than the double-trap device and glue-type sex lure device, respectively (Figure 3B). Analyzing the number of rice pests trapped, for *C. medinalis*, there was no significant difference in the amount of trapping between the double-trap device (29.33 ± 2.47) and glue-type sex lure device (28.33 ± 1.57) (*p* = 0.7609), but both were significantly higher than for the triple-trap device (10.67 ± 0.91) (Figure 3C). The numbers of rice leafhoppers (5.00 ± 0.77) and *C. suppressalis* (9.67 ± 2.70) trapped by the triple-trap device were higher than those of the double-trap device and glue-type sex lure device, and the number of *P. guttata* (6.67 ± 1.13) trapped by the triple-trap device was higher than that of the glue-type sex lure device (Figure 3C).

In field trial 2, the total capture rates of five types of natural enemies and other beneficial insects were higher for the triple-trap device (29.67 ± 1.16) and double-trap device (26.67 ± 1.64) than for the glue-type sex lure device (8.33 ± 1.93) (Figure 4A). Especially for *Coccinella*, the capture amounts of the triple-trap device (15.33 ± 1.39) and double-trap device (15.33 ± 0.40) were both higher than that for the glue-type sex lure device (2.67 ± 1.37); the number of Ichneumonidae (3.67 ± 0.33) and *Chrysopa* (2.33 ± 0.62) trapped by the triple-trap device was also higher than that of the glue-type sex lure device; the double-trap device also attracted more Ichneumonidae (5.33 ± 0.58) than the glue-type sex lure device (Figure 4B). The capture rates of non-target *Muscomorpha* of the triple-trap device (339.33 ± 19.43) and double-trapping device (316.00 ± 21.87) were both higher than those of the glue-type sex lure device (68.67 ± 11.05) (Figure 4C). The benefit–harm ratio (the ratio of the number of beneficial insects such as natural enemies captured to the number of rice pests captured) was highest for the triple-trap device and lowest for the glue-type sex lure device (Figure 4D).

### 3.3. Comparison of the Effects of Light-Transmitting Triple-Trap Device, Double-Trap Device, Light Green Sticky Plate Device, and Glue-Type Sex Lure Device

#### 3.3.1. Application of the sp(Cs)

In field trial 3, in terms of the relative abundance of trapped rice pests, *Empoasca flavescens* was the highest in both the triple-trap device and double-trap device, followed by *C. suppressalis*. In the light green sticky plate device, *E. flavescens* was the highest, followed by *Elophila fengwhanalis*. In the glue-type sex lure device, *C. suppressalis* was the highest, followed by *E. fengwhanalis* (Figure 5A). In terms of the number of rice pest species trapped, the triple-trap device (12 species) captured 10, 4, and 4 more species than the glue-type sex lure device, double-trap device, and light green sticky plate device, respectively (Figure 5B). Regarding the number of *C. suppressalis* attracted, those of the glue-type sex lure device (69.67 ± 5.81) and triple-trap device (49.67 ± 8.45) were both higher than for the light green sticky plate device (0.33 ± 15.63), and there was no significant difference among the glue-type sex lure device, triple-trap device, and double-trap device (36.67 ± 3.63) (Figure 5C). For the trapping of *S. inferens*, the triple-trap device (2.00 ± 0.75) scored higher than the other three devices. For the trapping of *Recilia dosalis*, the triple-trap device (1.67 ± 0.60) scored higher than the double-trap device and glue-type sex lure device. For the trapping of *E. flavescens*, the triple-trap device (109.67 ± 21.12) and double-trapping device (89.67 ± 10.97) both scored higher than the glue-type sex lure device, with the triple-trap device trapping more than the light green sticky plate device (53.67 ± 10.32). The number of *S. miscanthi* trapped by the triple-trap device (26.67 ± 6.21) and double-trap device (15.67 ± 2.20) was higher than that by glue-type sex lure device (Figure 5C). The order of the total number of rice pests trapped is as follows: triple-trap device > double-trap device > light green sticky plate device > glue-type sex lure device. The triple-trap device (211.00 ± 13.43) scored significantly higher than the light green sticky plate device (93.33 ± 17.07) and glue-type sex lure device (92.67 ± 14.51), and better than the double-trap device (152.00 ± 10.58). The double-trap device was better than the light green sticky plate device and glue-type sex lure device (Figure 5C). There was no significant difference in the number of insects captured by the four trapping devices for *S. furcifera*, *E. fengwhanalis*, *Typhlocybinae pomaria*, *N. aenescens*, *P.auttata*, *Lissorhoptrus oryzophilus*, *L. striatellus*, and *Nephotettix cincticeps* (Figure 5C). The dynamic trends in the capture of overwintering adult populations of *C. suppressalis* were generally consistent; the peak period of moth capture by the triple-trap device and double-trap device was the same (Figure 6).

In field trial 3, among the four devices, the triple-trap device captured the most non-rice pest species (pests that harm other crops) (10 species), specifically, 6, 6, and 4 more than the light green sticky plate device, glue-type sex lure device, and double-trap device, respectively (Figure 7B). In addition, the relative abundance of *Erythroneura melia* was highest in the triple-trap device, double-trap device, and light green sticky plate device, accounting for 91.99%, 89.86%, and 83.93%, respectively. The relative abundance of *Bactrocera tau* was highest in the glue-type sex lure device, accounting for 50.00%, followed by *Conogethes punctiferalis*, accounting for 25.00% (Figure 7A). The triple-trap device captured more of the other crop pests (non-rice pests) (104.00 ± 32.72) and more *Plutella xylostella* (2.33 ± 0.50) than the other three devices. It also captured more *E. melia* (95.67 ± 31.62) than the glue-type sex lure device and light green sticky plate device (15.67 ± 11.51) (Figure 7C). The light green sticky plate device captured the most fruit flies (2.00 ± 0.30), followed by the triple-trap device (1.67 ± 0.08). Both devices scored higher than the glue-type sex lure device and double-trap device (0.67 ± 0.08). However, the glue-type sex lure device captured more *C. punctiferalis* and *Leucania roseilinea* than the other three devices (Figure 7C). There was no significant difference in the number of traps used by the four devices for *L. roseilinea*, *Heliocoverpa assulta*, Noctuidae, *Melanotus caudex*, *Stephanitis nashi*, *Apolygus lucorum*, *Monolepta hieroglyphica*, and Tortricidae (Figure 7C).

In field trial 3, in terms of the relative abundance of trapped rice natural enemies and other beneficial insects, Ichneumonidae was the most abundant in the triple-trap device and double-trap device, accounting for 37.50% and 36.00%, respectively, followed by Araneae at 25.00% and 30.00%, respectively. The light green sticky plate device had the highest abundance of Ichneumonidae and Araneae, both accounting for 41.86%. The glue-type sex lure device had the highest abundance of Araneae, accounting for 68.42%, followed by Ichneumonidae, accounting for 15.79% (Figure 8A). Eight species of beneficial insects, including natural enemies, were trapped by the four types of traps. The triple-trap device and double-trap device attracted two and three more species of beneficial insects, respectively, than the light green sticky plate device and glue-type sex lure device (Figure 8B). The triple-trap device captured the most insects. The triple-trap device (18.67 ± 1.88) and double-trap device (16.67 ± 1.60) captured more insects than the glue-type sex lure device (6.33 ± 3.58). There was no significant difference between the triple-trap device and double-trap device or light green sticky plate device (14.33 ± 1.34) (Figure 9B). The double-trap device captured more *Coccinella* (3.67 ± 0.92) than the glue-type sex lure device (0.67 ± 0.08), and the triple-trap device captured more Staphylinidae (3.00 ± 1.04) than the glue-type sex lure device and light green sticky plate device. There was no statistically significant difference in the number of Araneae, *Chrysopa*, *Anisoptera*, Ichneumonidae, Cicindelidae, and *Apis* among the four types of devices (Figure 8C). Among the four types of devices, the light green sticky plate device captured the highest number of non-target *Muscomorpha* (53.00 ± 8.78), with no significant difference between the light green sticky plate device and triple-trap device (49.67 ± 6.47) and double-trap device (46.33 ± 3.63). All three devices captured more than the glue-type sex lure device (0.67 ± 3.68) (Figure 8D). The triple-trap device also captured more *Nilaparvata bakeri* and *Nilaparvata muiri* (19.00 ± 2.54) than lthe ight green sticky plate device (6.33 ± 1.67), double-trap device (4.67 ± 0.36), and glue-type sex lure device (Figure 8E).

Comprehensive analysis revealed that the triple-trap device (315.00 ± 43.33) captured a higher number of various crop pests than the glue-type sex lure device (95.33 ± 10.50), light green sticky plate device (112.00 ± 28.83), and double-trap device (175.00 ± 25.35) (Figure 9A). The benefit–harm ratio (the ratio of the total number of beneficial insects, including natural enemies, captured to the total number of crop pests captured) was lowest for the triple-trap device, and highest for the light green sticky plate device (Figure 9C).

#### 3.3.2. Application of sp(Cm)

In field trial 4, in terms of the relative abundance of major rice pests, the highest abundance among all devices was for *S. furcifera* (Figure 10A). The four trapping devices caught a total of 10 major pests, including *C. medinalis*, *S. furcifera*, and *C. suppressalis*. The triple-trap device attracted the most insect species (nine species), five species more than the double-trap device, five species more than the light green sticky plate device, and seven species more than the glue-type sex lure device (Figure 10B). It also captured the largest number of pests (60.33 ± 11.04), outperforming the glue-type sex lure device (12.67 ± 2.85) and double-trap device (20.67 ± 3.39) (Figure 10C). The glue-type sex lure device captured the highest number of *C. medinalis* (3.33 ± 0.14), which was higher than for the light green sticky plate device (0.33 ± 0.14), triple-trap device (0.67 ± 0.46), and double-trap device (1.00 ± 0.74) (Figure 10C). The triple-trap device captured the highest number of *S. furcifera* (39.67 ± 6.41), which was higher than for the glue-type sex lure device (9.33 ± 1.09), and had no significant difference with the double-trap device (18.67 ± 2.32) or glue-type sex lure device (22.33 ± 7.50) (Figure 10C). For *L. striatellus*, the triple-trap device captured a higher number (3.33 ± 0.58) than the glue-type sex lure device and double-trap device (0.67 ± 0.63), and the light green sticky plate device (1.67 ± 0.38) captured more than the glue-type sex lure device (Figure 10C). The triple-trap device captured a higher number of *C. suppressalis* (8.67 ± 0.90) than the other three devices (Figure 10C). For rice leafhoppers (total amount of three types of leafhoppers), the triple-trap device (6.33 ± 3.91) produced the highest capture rate, surpassing that of the other three devices, followed by the double-trap device (0.33 ± 1.08). For the total capture rate of *N. bakeri* and *N. muiri*, the triple-trap device (5.00 ± 0.52) produced the highest capture rate, surpassing the glue-type sex lure device, double-trap device (0.67 ± 0.46), and light green sticky plate device (1.33 ± 0.22) (Figure 10C). According to the continuous trapping of *S. furcifera* using four types of devices from April to August, the main peak periods of the four types of devices tended to be consistent in early to mid-July. The main peak periods of the triple-trap device, double-trap device, and light green sticky plate device coincided in early August. The peak period of each generation was highest with the triple-trap device, and the insects appeared earliest in the year (Figure 11).

### 3.4. Comparison of Effects of Light-Shielding and Light-Transmitting Triple-Trap Device

#### 3.4.1. Application of sp(Cm)

In field trial 5, the light-transmitting triple-trap device trapped 11 rice pest species, 6 species more than the light-shielding triple-trap device; the total number of insects trapped (273.33 ± 19.70) was greater than that with the light-shielding device (150.33 ± 20.51) (*p* = 0.0124); the number of *Recilia dosalis* trapped was greater with the light-transmitting device (245.33 ± 12.72) than with the light-shielding device (141.00 ± 20.21) (*p* = 0.0120) (Figure 12A). However, the number of *C. medinalis* trapped was less with the light-transmitting device (0.33 ± 0.33) than with the light-shielding device (2.67 ± 0.67) (*p* = 0.0352) (Figure 12A). There was no significant difference in the capture rates of other pests between the two types of triple-trap devices. However, the light-transmitting triple-trap device increased the capture rates of *C. suppressalis*, *N. aenescens*, *C. exigua*, *L. striatellus*, *Nilaparvata lugens*, *S. furcifera*, *S. inferens*, *P. auttata*, and Pentatomidae. For example, the capture rate of *C. suppressalis* (10.67 ± 4.06) increased by 9.67 times compared with that of the light-shielding type (1.00 ± 0.58), and the capture rate of *S. furcifera* (7.00 ± 2.08) increased by 31.33% compared with that of the light-shielding type (5.33 ± 0.67) (Figure 12A). At the same time, the number of non-target *Muscomorpha* trapped by the light-transmitting device (49.67 ± 2.60) was lower than that by the light-shielding device (134.00 ± 13.20) (*p* = 0.0033) (Figure 12B). The light-shielding device captured one more natural enemy species (four species) than the light-transmitting device, while the total number of insects trapped did not differ significantly between the two types of devices (*p* = 0.8149) (Figure 12A,D). The light-transmitting device captured more Staphylinidae (4.33 ± 0.33) than the light-blocking device (2.00 ± 0.58) (*p* = 0.0249) (Figure 12C). The benefit–harm ratio of the light-transmitting device was lower than that of the light-shielding device (Figure 12D).

#### 3.4.2. Application of sp(Cs)

In field trial 6, the light-transmitting triple-trap devices trapped 11 species of rice pests, 2 more than the light-shielding type (Figure 13A). The total number of insects trapped (154.67 ± 3.76) was greater than that of the light-shielding type (77.33 ± 3.76) (*p* = 0.0015), among which the number of *R. dosalis* trapped (133.00 ± 5.51) was greater than that of the light-shielding type (66.33 ± 10.37) (*p* = 0.0047) (Figure 13A). As a result, the total number of rice leafhoppers trapped, mainly electric leafhoppers, (137.67 ± 5.84), was also greater than that with the light-blocking-type device (70.00 ± 10.26) (*p* = 0.0046) (Figure 13B,C). The light-transmitting triple-trap devices had no significant difference in attracting other insect species, but increased the capture of *C. exigua*, *C. suppressalis*, and *N. aenescens*; for example, the capture of *C. exigua* (3.67 ± 1.45) was 10.12 times higher than that of the light-shielding type (0.33 ± 0.33); the light-transmitting type captured *C. suppressalis* (3.00 ± 1.53) and *N. aenescens* (1.00 ± 0.58), while the light-shielding-type device did not capture these two pests (Figure 13A). Regarding the trapping of beneficial insects such as natural enemies, the light-transmitting triple-trap devices captured more Staphylinidae (22.67 ± 4.26) than the light-shielding type (10.00 ± 1.53) (*p* = 0.0487) (Figure 13D). The total number of traps captured by the light-transmitting type (28.33 ± 5.90) was not significantly different from that of the light-shielding type (17.67 ± 3.84) (*p* = 0.2043), and its benefit–harm ratio was lower than that of the light-shielding type (Figure 13D,F). The number of non-target *Muscomorpha* captured by the light-transmitting type (30.00 ± 4.93) was lower than that by the light-shielding type (200.00 ± 18.23) (*p* = 0.0008) (Figure 13E).

## 4. Discussion

### 4.1. Comparison of Light-Shielding Triple-Trap Device, Double-Trap Device, and Glue-Type Sex Lure Device

In field trial 1 and field trial 2, when *C. medinalis* was prevalent, the use of its insect sex pheromones for trapping showed that the number of *C. medinalis* trapped by the light-shielding double-trap device was not significantly different from that of the control glue-type sex lure device. At the same time, it would increase the trapping of pests such as *P. auttata* and rice planthoppers and would also significantly increase the attraction of natural enemy insects such as *Coccinella* and Ichneumonidae and non-target *Muscomorpha*, but the benefit–harm ratio of the two types of devices was similar. The light-shielding triple-trap device increased the number of rice pest species trapped compared to the control glue-type sex lure device; significantly increased the capture of rice leafhoppers, *Coccinella*, natural enemy insects, and non-target *Muscomorpha*; and significantly increased the capture rate of *C. suppressalis*, *P. auttata*, and its natural enemies, *Anisoptera* and Ichneumonidae, but significantly reduced the capture of *C. medinalis* (reduced by 62.34%) and the benefit–harm ratio was high. The light-shielding triple-trap device adds ultraviolet light to the light-shielding double-trap device; this change increases the number of rice pest species trapped, significantly increases the capture of rice leafhoppers, and significantly increases the capture of *S. furcifera* (increased by 1.35 times) and the capture of *C. suppressalis*, but it will significantly reduce the capture of *C. medinalis* (reduced by 63.62~66.71%); in addition, this change makes the population dynamics and peak period of *S. furcifera* adults more obvious.

### 4.2. Comparison of the Light-Transmitting Triple-Trap Device, the Double-Trap Device, the Light Green Sticky Plate Device, and the Glue-Type Sex Lure Device

In field trial 3, in trapping the overwintering adults of *C. suppressalis*, there was no significant difference among the triple-trap device, double-trap device, and control glue-type sex lure device, and the dynamic trends of the trapped populations were generally consistent, but the peak period of moths trapped was slightly delayed compared with the control glue-type sex lure device. Preliminary analysis showed that this might be due to the tendency of female *C. suppressalis* moths to prefer light green and their egg-laying and breeding characteristics in the field, which led to an increase in the number of female moths trapped on the color plate and a delay in the peak period. The 2024 experimental investigation found that on 12 May, 12 July, and 12 August, which were the peak periods of each generation of stem borer and the peak egg-laying period, trapped female *C. suppressalis* and their eggs were found on the triple-trap device (Figure 14). Studies have reported that the number of female moths of *C. suppressalis* trapped through the combination of sex attractants + black light lamps and ultraviolet light lamps (365 nm) was significantly higher than that through the combination of sex attractants and yellow boards and green boards [31,32]; this phenomenon is similar to that observed in this study. Therefore, further research is necessary on the use of the triple-trap device to trap female moths of *C. suppressalis* and on the male-to-female ratio. The triple-trap device captured the most rice pests and other crop pests, the most beneficial insects, and the most total insects; it also captured the most early-stage *S. miscanthi*, rice leafhoppers, *S. inferens*, and *S. furcifera*. Furthermore, the total number of rice pests captured was significantly higher than that with the control glue-type sex lure device and light green sticky plate device, and significantly higher than that with the double-trap device. The number of other crop pests captured was significantly higher than for the other three types of devices. Among the four devices, the triple-trap device had the lowest benefit-to-harm ratio. Compared with the control (glue-type sex lure device), triple-trap device, double-trap device and light green sticky plate device all significantly increased the capture of non-target flies (*Muscomorpha*).

In field trial 4, when *C. medinalis* was prevalent, the results of sp(Cm) showed that the control glue-type sex lure device had the highest capture rate, which was significantly higher than that of the light green sticky plate device and significantly higher than that of the double-trap device and triple-trap device (Figure 10C). The triple-trap device has a good effect on the simultaneous trapping of other major rice pests and captures the largest number of insect species; it has the highest total capture volume of the following: *C. suppressalis*, *S. furcifera*, *L. striatellus*, *S. inferens*, rice leafhopper, *C. exigua*, rice planthopper, and total capture volume of major rice pests. Compared with the other three devices, it significantly increased the capture volume of striped *C. suppressalis* to an extreme degree and significantly increased the capture volume of *S. inferens*. When there is a light outbreak of *S. furcifera* (Figure 11), the other three devices, except the glue-type sex lure device, can trap the first four (second) generations of the main pest in early to mid-July [11,28], and the main peak period of trapping the fifth (third) generation of the main pest in early August is more obvious and coincides. The triple-trap device can detect insects the earliest and has the highest trapping number for each generation peak period, which is conducive to the early detection of insect infestation.

### 4.3. Comparison of Light-Transmitting and Light-Shielding Trapping of Triple-Trap Device

In field trial 5 and field trial 6, using either sp(Cm) or sp(Cs), the light-transmitting triple-trap device captured two to six more rice pest species than the light-shielding triple-trap device, significantly or extremely significantly increasing the total capture of rice pests (by 0.82–1.00 times) and the capture of *R. dosalis* (by 0.74–1.01 times). Furthermore, under the same conditions, the light-transmitting-type device captured *S. furcifera*, *C. suppressalis*, rice leafhoppers, *L. striatellus*, and *C. exigua* by 0.31 times, 9.67 times, 0.74–0.97 times, 2.03 times, and 10.12 times, respectively. However, when there was a mild outbreak of *C. medinalis*, the capture amount of the light-transmitting-type device was significantly lower than that of the light-shielding-type device (reduced by 87.64%), and the capture amount of Staphylinidae was significantly increased, resulting in a significant decrease in the benefit–harm ratio, but the number of non-target *Muscomorpha* captured showed an extremely significant decrease, and the interference was relatively small.

### 4.4. Selection of Rice Pest Trapping Devices

Under the same conditions, the number of major rice pests such as *S. furcifera*, *C. medinalis*, rice leafhopper, and *C. suppressalis* trapped by different trapping devices showed significant or obvious changes. The results show that when the population density of the two migratory pests was high, adding ultraviolet light significantly increased the capture rate of *S. furcifera*, changing the transparency of the cover further increased the capture rate, and removing the ultraviolet light significantly reduced the capture rate. This indicates that ultraviolet light and transparent light environments have a significant and obvious positive attraction and synergistic effect on the attraction of *S. furcifera*, respectively (Figure 15A). Furthermore, for *C. medinalis*, adding UV light significantly decreased the capture rate, changing the transparency of the cover caused a further significant decrease in the capture rate, and removing UV light significantly increased the capture rate. This indicates that UV light and a transparent light environment have a significantly negative and significant phototaxis-weakening effect on the capture of *C. medinalis*, respectively (Figure 15B). For rice leafhoppers, mainly *R. dosalis*, the results also show that ultraviolet light and transparent light environments had extremely significant or significantly positive attraction and synergistic effects on their attraction, respectively (Figure 16). In the absence of sp(Cs), violet light and a transparent light environment also had highly significant and significantly positive trend attraction and synergistic effects (Figure 17), respectively; when using sp(Cs), changing the transparency of the cover significantly increased the number of *C. suppressalis* trapped. Whether violet light was added or not, there was no significant difference compared to the single-lure method (the light green sticky plate device). However, the number of traps using violet light was significantly higher than that of the light green sticky plate lure method without violet light. This indicates that while sp(Cs) is dominant, violet light and a transparent light environment also have significant positive trend attraction and synergistic effects.

The translucent three-combination trapping method, based on a combination of color plates, insect sexual information, insect trap lights, and translucent covers, combined with a combination of purple light traps, translucent covers, and a preferred light green plate, along with insect pheromones, simultaneously attracted major pests such as *C. suppressalis*, rice planthoppers, rice leafhoppers, *S. inferens*, *S. miscanthi*, and other crops, as well as *Plutella xylostella*. This significantly increased the variety and number of trapped pests, achieving the lowest benefit-to-harm ratio (0.059) compared to the simultaneously tested glue-type sex lure device and a lower benefit-harm ratio (1:13.94) than the insect-trapping lamp with a wavelength of 365nm [33,34]. The damage to the natural enemy insects and other beneficial insects was relatively small, and the non-target insects had little interference with them. At the same time, the ultraviolet lamp and translucent cover in the device showed an extremely significant or significantly negative growth effect on the number of *C. medinalis* trapped; the experiment confirmed that *C. medinalis* has a negative attraction characteristic to ultraviolet light waves and translucent covers. Therefore, in rice production, the light-shielding double-trap device or glue-type sex lure device are recommended for monitoring *C. medinalis*, while the light-transmitting triple-trap device is recommended for monitoring and controlling other major pests besides *C. medinalis*.

The phototactic response of insects varies depending on the insect species, light properties, and physiological state of the insects [35,36,37]. Indoor studies have shown that light waves have a significant effect on the phototactic rate of *S. furcifera*. *S. furcifera* are more sensitive to blue (470 nm) and green (515~550 nm) light, while green light has a relatively low attraction rate to their natural enemy, *Cyrtorhinus lividipennis* [38]. There is no significant difference in the capture rate of *C. suppressalis* and planthoppers when insect sex pheromones are combined with black light or ultraviolet light of a wavelength of 365 nm [31]. Insects have a specific selectivity and preferences for different colors, and also have different preferences for different gradients of the same color [39,40,41,42,43]. Therefore, on the basis of strengthening the research on the physical and chemical characteristics of insect tropism, we should give full play to and utilize the synergistic and complementary advantages of different mechanism combinations such as color plates, insect sex pheromones, specific lights of different wavelengths, and light-shielding (light-transmitting) covers, so as to maximize their strengths and minimize their weaknesses, as well as further develop multi-source physical and chemical attractant combination devices suitable for different crops and pest species, providing simple or intelligent tools and equipment for the accurate and efficient monitoring and control of insect infestations.

## 5. Conclusions

In rice production, the light-shielding double-trap device, which uses sex pheromones and light green sticky plate, is the preferred method for trapping and monitoring *C. medinalis*. For other major rice pests besides *C. medinalis*, the light-transmitting triple-trap device is the preferred method for simultaneous trapping and monitoring. This study further reveals the synergistic effects of trapping methods based on different combinations of color panels, insect sex pheromones, light sources, and light-shielding (light-transmitting) covers on rice pests, providing a scientific basis for the rational and effective application of different multi-source physical and chemical trapping methods for pest monitoring and control.

## Figures and Tables

**Figure 1 insects-16-01001-f001:**
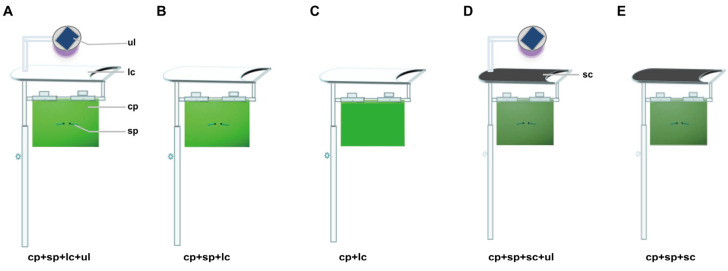
Different types of pest trap devices. Light-transmitting triple-trap device (**A**); light-transmitting double-trap device (**B**); light-transmitting light green sticky plate device (**C**); light-shielding triple-trap device (**D**); light-shielding double-trap device (**E**).

**Figure 2 insects-16-01001-f002:**
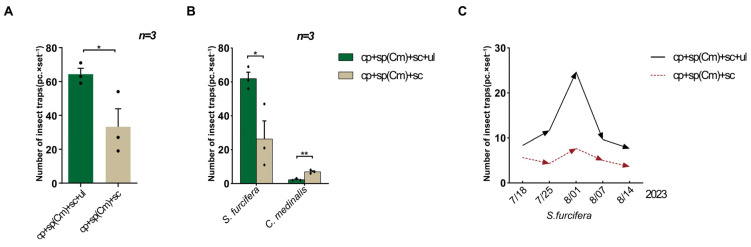
Comparison of light-shielding triple-trap device and double-trap device using sp(Cm). Field trial 1: comparison of total captures of two migratory pests of rice (**A**); comparison of the trapped quantity of *S. furcifera* and *C. medinalis* (**B**); population dynamics of trapped *S. furcifera* adults (**C**). Data are mean ± standard error; * and ** indicate the significance of the difference at the 0.05 and 0.01 levels, respectively.

**Figure 3 insects-16-01001-f003:**
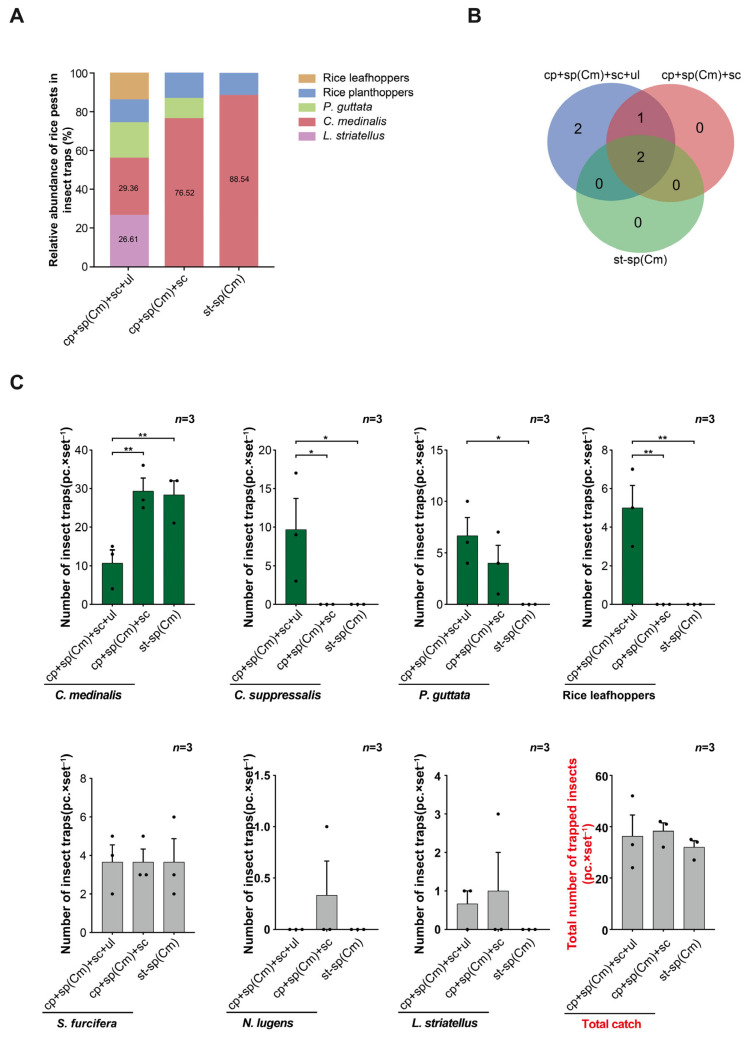
Comparison of the light-shielding triple-trap device, double-trap device, and glue-type sex lure device using the sp(Cm). Field trial 2: relative abundance of trapped rice pests (%) (**A**); number of trapped rice pest species (**B**); comparison of the effects of luring several major rice pests (**C**). Data are mean ± standard error; * and ** indicate significant differences (*p* < 0.05) and extremely significant differences (*p* < 0.01), respectively.

**Figure 4 insects-16-01001-f004:**
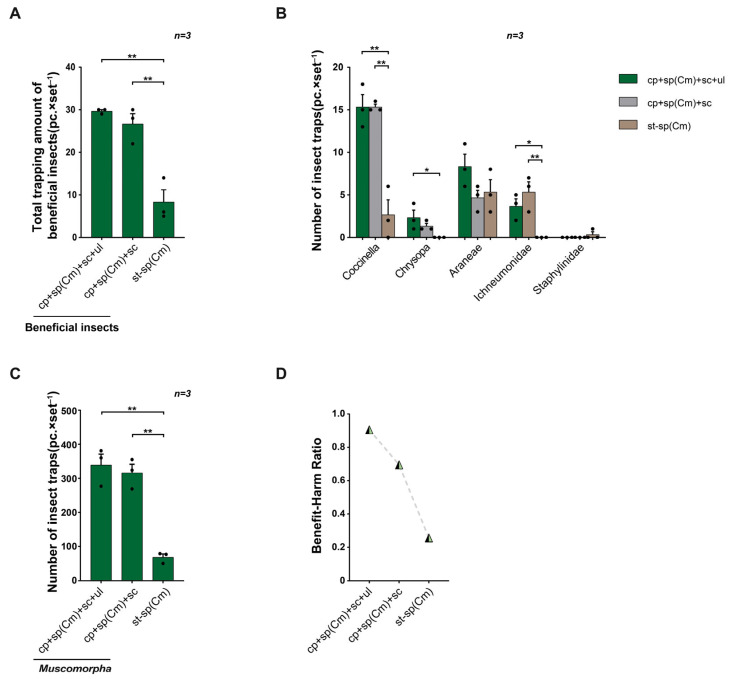
Comparison of the light-shielding triple-trap device, double-trap device, and glue-type sex lure device using the sp(Cm). Field trial 2: comparison of the total number of natural enemies and beneficial insects trapped by the three devices (**A**); comparison of the number of different types of natural enemies and other beneficial insects trapped by the three devices (**B**); comparison of the number of non-target *Muscomorpha* trapped by the three devices (**C**); the benefit–harm ratio of the three devices (**D**). Data are mean ± standard error; * and ** indicate significant difference (*p* < 0.05) and extremely significant difference (*p* < 0.01), respectively.

**Figure 5 insects-16-01001-f005:**
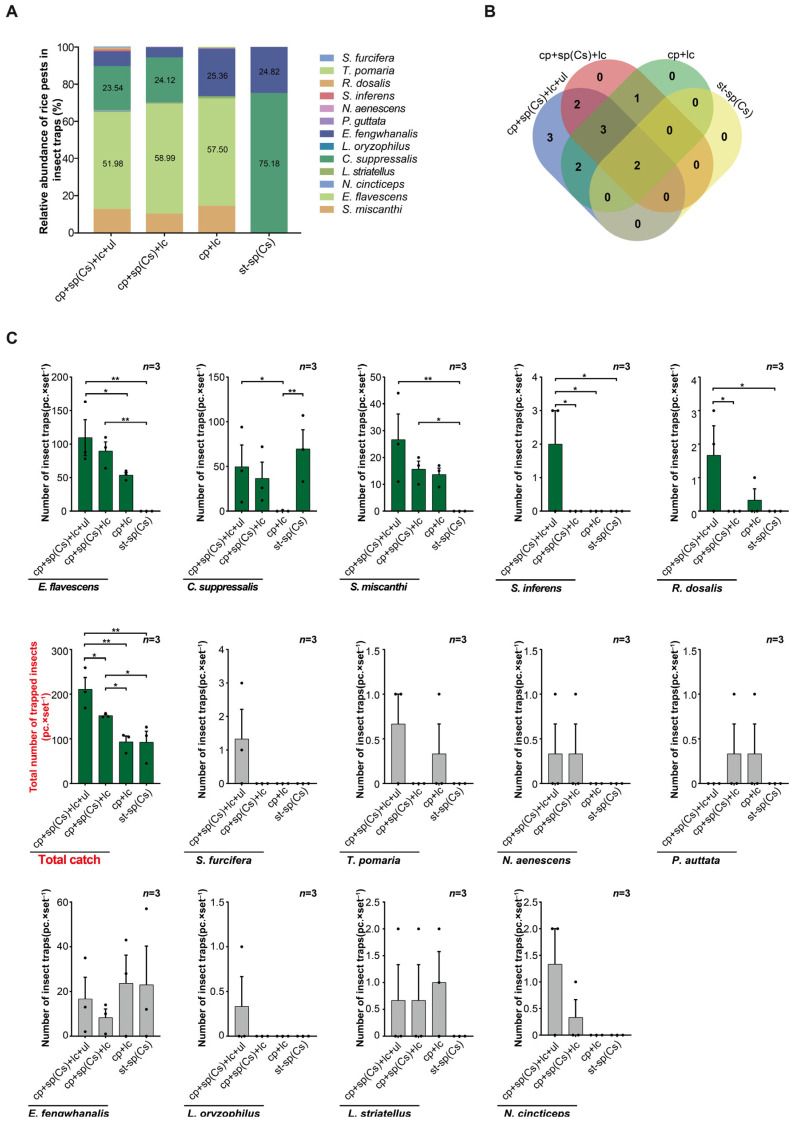
Comparison of light-transmitting triple-trap device, double-trap device, light green sticky plate device, and glue-type sex lure device for luring major rice pests. Field trial 3: relative abundance of lured rice pests (**A**); number of lured rice pest species (**B**); comparison of several major lured rice pests (**C**). Data are mean ± SD; * and ** indicate significant difference (*p* < 0.05) and extremely significant difference (*p* < 0.01), respectively.

**Figure 6 insects-16-01001-f006:**
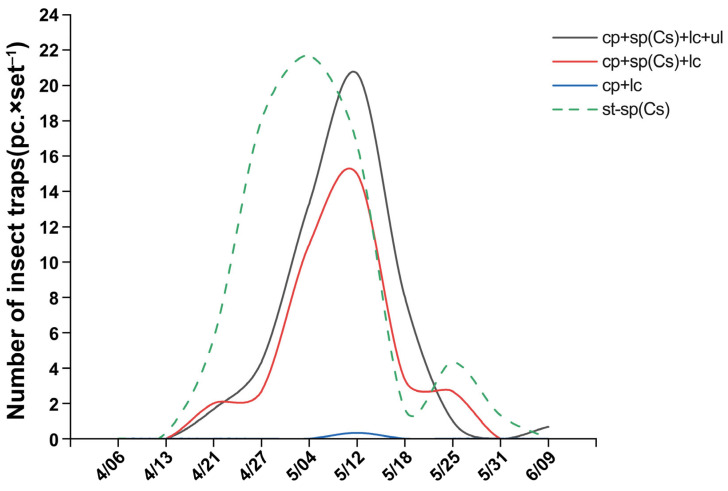
Comparison of the dynamics of overwintering adults of *C. suppressalis* using a light-transmitting triple-trap device, a double-trap device, a light green sticky plate device, and a glue-type sex lure device in field trial 3.

**Figure 7 insects-16-01001-f007:**
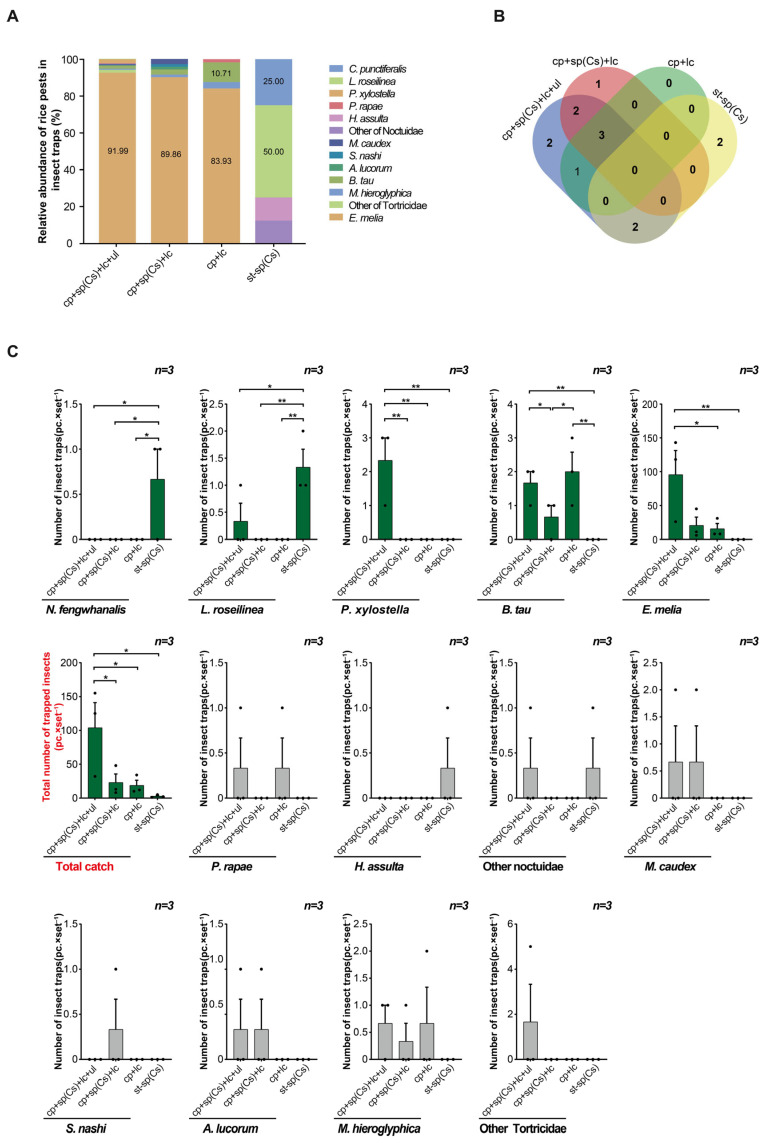
Comparison of light-transmitting triple-trap device, double-trap device, light green sticky plate device, and glue-type sex lure device for trapping other crop pests. Field trial 3: relative abundance of other crop pests in the traps, which harm other crops but not rice (**A**); number of other pest species lured (**B**); comparison of several other pests lured (**C**). Data are mean ± SD; * and ** indicate significant difference (*p* < 0.05) and extremely significant difference (*p* < 0.01), respectively.

**Figure 8 insects-16-01001-f008:**
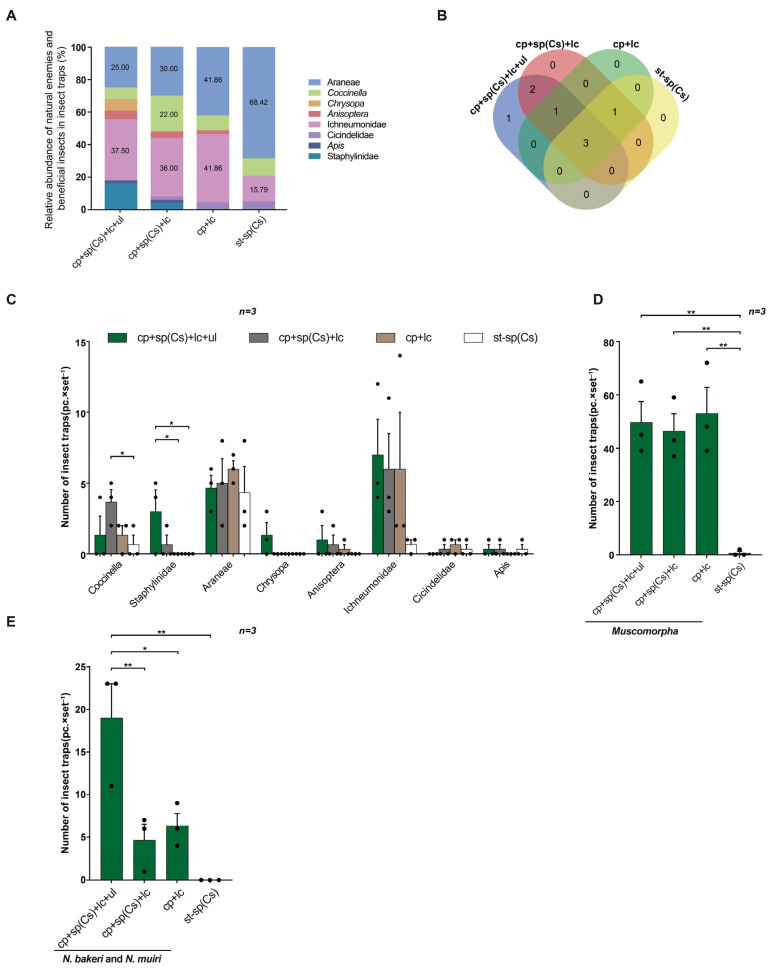
Comparison of light-transmitting triple-trap device, double-trap device, light green sticky plate device, and glue-type sex lure device for luring natural enemies, beneficial insects, and non-target insects. Field trial 3: relative abundance of natural enemy beneficial insects attracted (**A**); types of natural enemy beneficial insects attracted (**B**); comparison of natural enemy beneficial insects attracted (**C**); non-target *Muscomorpha* attracted (**D**); total number of *N. bakeri* and *N. muiri* trapped (**E**). Data are mean ± standard error; * and ** indicate significant differences at the 0.05 and 0.01 levels, respectively.

**Figure 9 insects-16-01001-f009:**
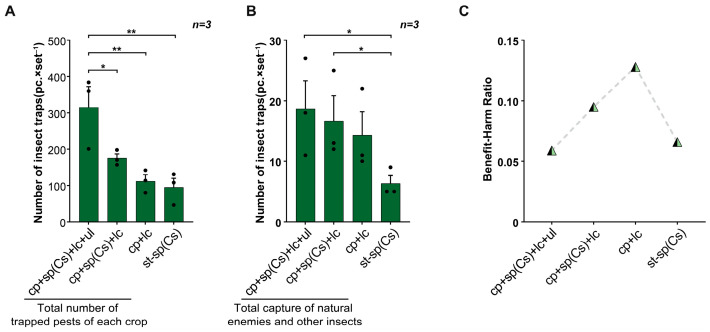
Comparison of the benefit–harm ratio of the light-transmitting triple-trap device, the double-trap device, the light green sticky plate device, and the glue-type sex lure device. Field trial 3: total number of pests attracted for each crop (rice and other surrounding crops) (**A**); total number of natural enemies and other insects attracted (**B**); benefit–harm ratio in the four devices (**C**). Data are mean ± SD; * and ** indicate significant difference (*p* < 0.05) and extremely significant difference (*p* < 0.01), respectively.

**Figure 10 insects-16-01001-f010:**
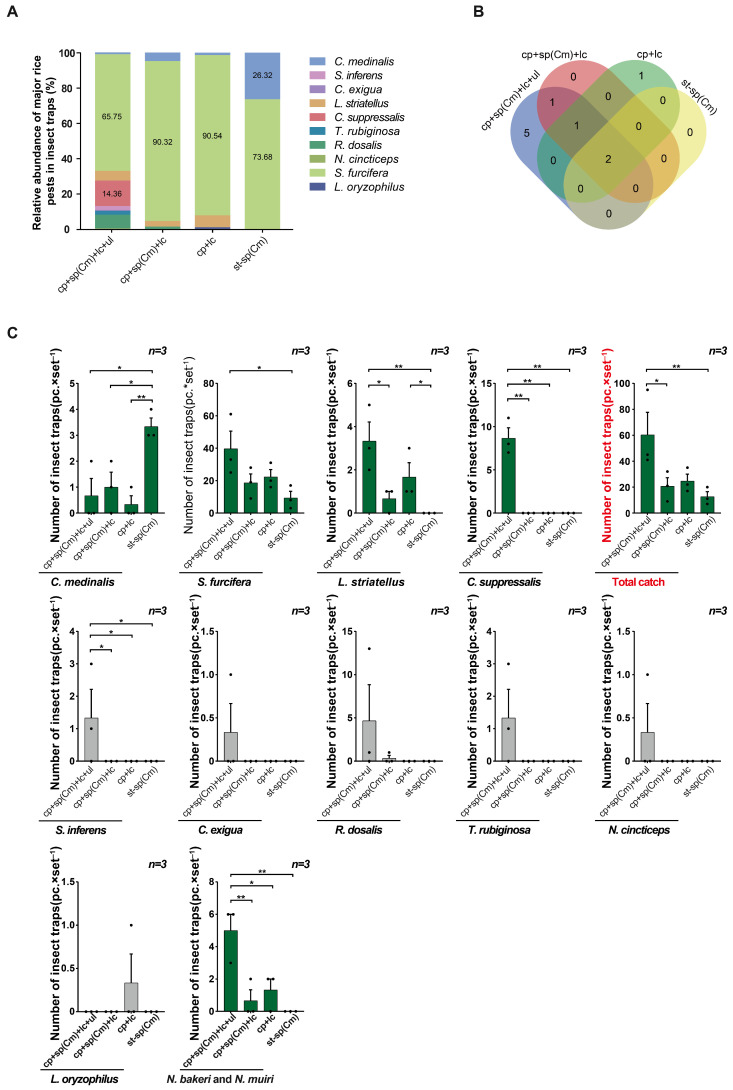
Comparison of the effects of the light-transmitting triple-trap device, the double-trap device, the light green sticky plate device, and the glue-type sex lure device. Field trial 4: relative abundance of various rice pests in four types of traps (**A**); number of rice pest species in four types of traps (**B**); comparison of the capture volume of major pests (**C**). Data are mean ± SE; * and ** indicate significant difference (*p* < 0.05) and extremely significant difference (*p* < 0.01), respectively.

**Figure 11 insects-16-01001-f011:**
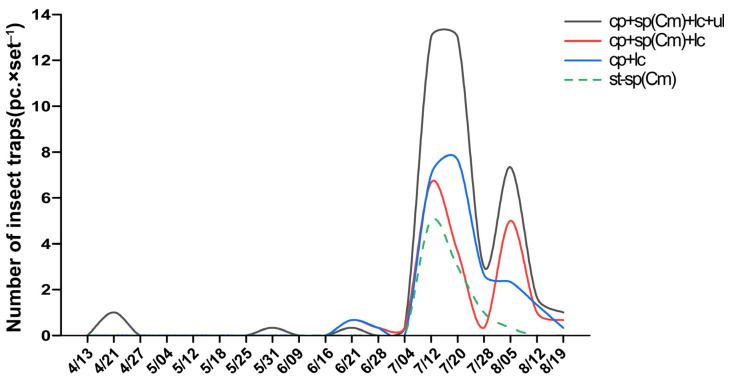
Comparison of light-transmitting triple-trap device, double-trap device, light green sticky plate device, and glue-type sex lure device in detecting the population dynamics of *S. furcifera* adults in field trial 4.

**Figure 12 insects-16-01001-f012:**
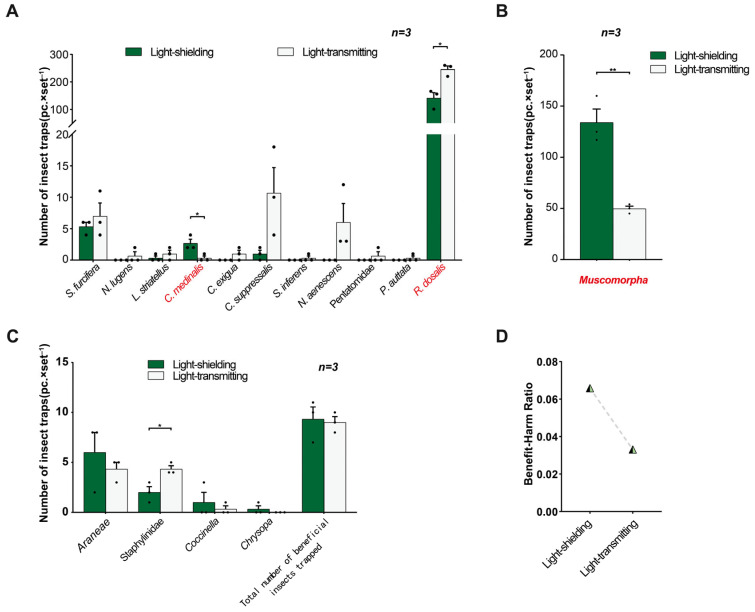
Comparison of the trapping effects of the light-shielding and light-transmitting triple-trap devices (using sp(Cm)). Field trial 5: comparative trapping of several major rice pests (**A**); comparative trapping of non-target *Muscomorpha* (**B**); comparative trapping of natural enemies and other beneficial insects (**C**); benefit–harm ratio (**D**). Data are mean ± standard error; * and ** indicate significant differences (*p* < 0.05) and extremely significant differences (*p* < 0.01), respectively.

**Figure 13 insects-16-01001-f013:**
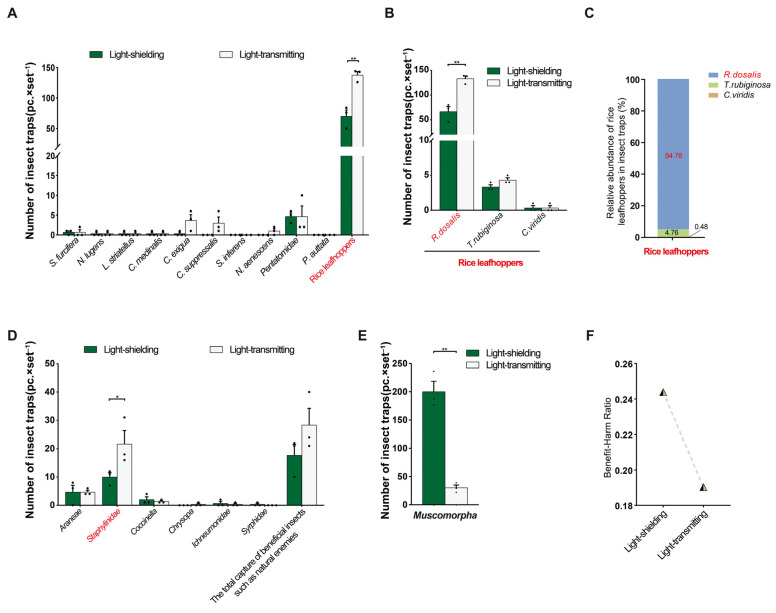
Comparison of effects of light-shielding and light-transmitting trapping devices (using sp(Cs)). Field trial 6: comparative trapping of several major rice pests (**A**); comparative trapping of rice leafhoppers (*R. dosalis*, *T. pomeria*, and *Cicadella viridis*) (**B**,**C**); comparative trapping of natural enemy beneficial insects (**D**); comparative trapping of non-target *Muscomorpha* (**E**); benefit–harm ratio (**F**). * and ** indicate significant difference (*p* < 0.05) and extremely significant difference (*p* < 0.01), respectively.

**Figure 14 insects-16-01001-f014:**
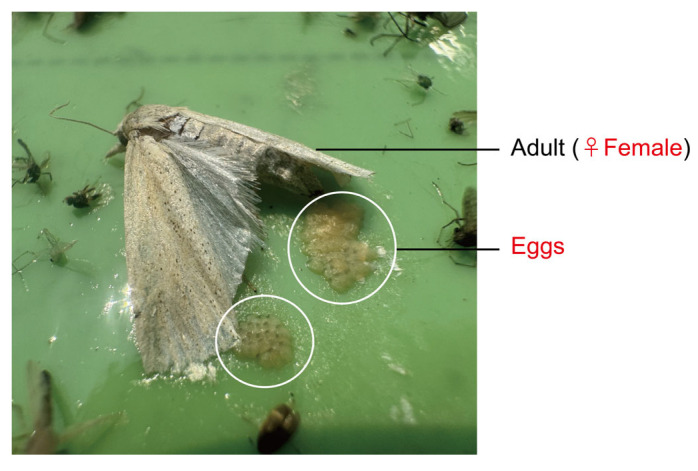
Female *C. suppressalis* and their eggs trapped with the light-transmitting triple-trap device.

**Figure 15 insects-16-01001-f015:**
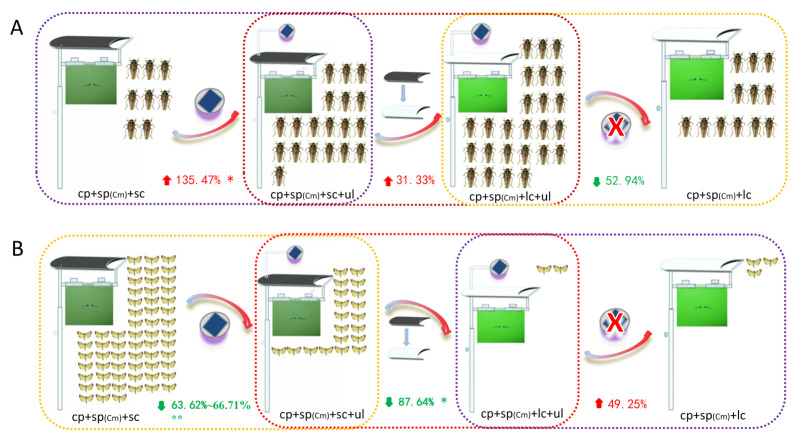
Schematic of the ratio of the amount of attractants used by different multi-source physical and chemical attractant control devices for “two migratory” rice pests: *S. furcifera* (**A**); *C. medinalis* (**B**). * and ** indicate significant differences (*p* < 0.05) and extremely significant differences (*p* < 0.01), respectively. ‘↑ (red)’and ‘↓ (green)’ indicate increase ana decrease, respectively. ‘× (red)’ means to delete the ultraviolet lamps.

**Figure 16 insects-16-01001-f016:**
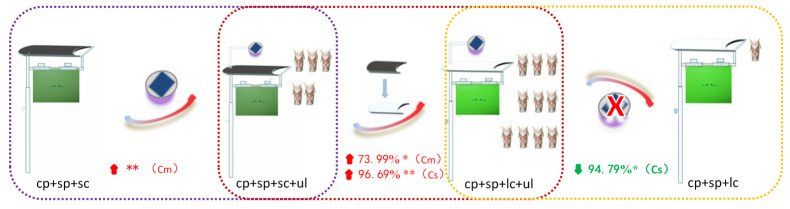
Schematic of the ratio of the number of rice leafhoppers attracted by different devices. * and ** indicate significant differences (*p* < 0.05) and extremely significant differences (*p* < 0.01), respectively. ‘↑ (red)’and ‘↓ (green)’ indicate increase ana decrease, respectively. ‘× (red)’ means to delete the ultraviolet lamps.

**Figure 17 insects-16-01001-f017:**
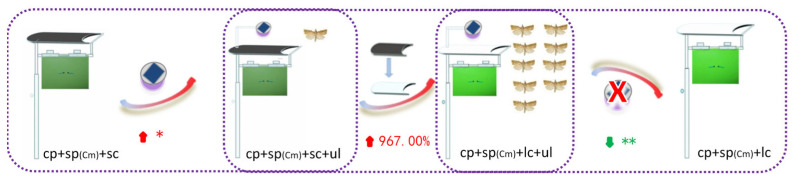
Schematic of the changes in the capture rate of *C. suppressalis* by installing sp(Cm) on different devices. * and ** indicate significant differences (*p* < 0.05) and extremely significant differences (*p* < 0.01), respectively. ‘↑ (red)’and ‘↓ (green)’ indicate increase ana decrease, respectively. ‘× (red)’ means to delete the ultraviolet lamps.

**Table 1 insects-16-01001-t001:** Experimental design of different types of devices, main objects of conducted survey, etc.

Field Trial	Year	Test Time	Rice Growing Period	Place	Lure Type	Different Types of Devices for Testing	Simultaneous Comparative Entrapment Survey Main Object
No. 1	2023	13 July–21 August	From the end of booting stage to waxy stage	Test site 1	*C. medinalis* (sp(Cm))	Light-shielding triple-trap device (cp + sp(Cm) + sc + ul) Light-shielding double-trap device (cp + sp(Cm) + sc)	Two migration rice pests (*C. medinalis* and *S. furcifera*)
No. 2		1–23 August	Jointing to late booting stage	Test site 2		Light-shielding triple-trap device (cp + sp(Cm) + sc + ul) Light-shielding double-trap device (cp + sp(Cm) + sc) Glue-type sex lure device (st-sp(Cm))	Major pests in the middle and late stages of rice production, beneficial insects such as natural enemies and major non-target *Muscomorpha*.
No. 3	2024	13 April–9 June	Seedling to jointing stage	Test site 1	*C. suppressalis* (sp(Cs))	Light-transmitting triple-trap device (cp + sp + lc + ul) Light-transmitting double-trap device (cp + sp + lc) Light-transmitting light green sticky plate device (cp + lc) Glue-type sex lure device (st-sp)	Overwintering rice pests such as *C. suppressalis*, other crop pests, beneficial insects such as natural enemy insects, and major non-target *Muscomorpha*.
No. 4		9 June–19 August	Jointing to yellow ripening stage		*C. medinalis* (sp(Cm))		“Two migration” rice pests and other major pests in the middle and late stages
No. 5		28 July–21 August	Jointing to grouting period	Test site 2	*C. medinalis* (sp(Cm))	Light-shielding triple-trap device (cp + sp + sc + ul) Light-transmitting triple-trap device (cp + sp + lc + ul)	“Two-migratory” rice pests and other major mid- and late-stage pests, beneficial insects such as natural enemy insects, and major non-target *Muscomorpha*
No. 6		21 August–6 September	From grain filling to yellow maturity stage		*C. suppressalis* (sp(Cs))		

## Data Availability

The original contributions presented in this study are included in the article. Further inquiries can be directed to the corresponding authors.

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
