# Peer review of "Synergistic Attraction and Ecological Effects of Multi-Source Physical and Chemical Trapping Methods with Different Mechanism Combinations on Rice Pests"

_insects, 2025, doi:10.3390/insects16101001_

Round 1
Reviewer 1 Report
Comments and Suggestions for Authors
This manuscript “Study on the Effect of Synchronous Trapping and Monitoring 2
of Rice Pests by Different Types of Multi-Source Physical and 3 Chemical Trapping Devices”
addresses an interesting issue related to the possibility of developing appropriate monitoring tools through the comparison of different methods and technologies.
As general comment, effective monitoring of crop-damaging species under consideration is achieved with simple, effective, and sensitive methodologies that respond to operational environmental conditions.
The work under review takes into consideration a large number of species and a great variety of variables which, when aggregated together, make any form of analysis very difficult.The authors' approach to the work is more experimental than scientific. For this reason, the paper lists primarily numerical results, sometimes repetitive, both in written form and in schematic representations of graphs and figures.In my opinion, the manuscript needs to be revised by eliminating the introductory and overly general sections. It is also necessary to avoid repetitions between the written part and the illustrations.Focus attention on the objectives of the activities in favor of simple, field-applicable, cost-effective and effective indications.In the attached PDF I tried to highlight some parts that should be shortened (in green) or that are the real scientific content (in red) that require a more in-depth reflection.

Author Response
|
Reply to reviewer 1 |
||||
|
1. Summary |
|
|
||
|
Dear reviewer, thank you very much for taking the time to review this manuscript and for your valuable feedback. Your suggestions have been very helpful in revising and improving the manuscript. We have carefully considered and revised your valuable comments. Please read our detailed response below. |
||||
|
2. Questions for General Evaluation |
Reviewer’s Evaluation |
Response and Revisions |
||
|
Does the introduction provide sufficient background and include all relevant references? |
Yes/Can be improved/Must be improved/Not applicable |
We have added more literature related to this study as references to increase the importance, necessity, and scientificity of this experimental research topic (in green). |
||
|
Is the research design appropriate? |
Yes/Can be improved/Must be improved/Not applicable |
|
||
|
Are the methods adequately described? |
Yes/Can be improved/Must be improved/Not applicable |
|
||
|
Are the results clearly presented? |
Yes/Can be improved/Must be improved/Not applicable |
We have revised the written results to reduce irrelevant repetitions that are clearly presented in the figures (in red). |
||
|
Are the conclusions supported by the results? |
Yes/Can be improved/Must be improved/Not applicable |
We have optimized the presentation of our findings to reduce confusion. |
||
|
Are all figures and tables clear and well-presented? |
Yes/Can be improved/Must be improved/Not applicable |
We have modified the table appropriately and increased the ppi value of the figures to more clearly illustrate the experimental design and the differences in experimental treatment combinations in different experiments (in yellow).
|
||
|
3. Point-by-point response to Comments and Suggestions for Authors |
||||
|
Comments 1: [The authors' approach to the work is more experimental than scientific.] |
||||
|
Response 1: Thank you for pointing this out. We agree with this comment. The purpose of our field trial research is to compare the effectiveness of different devices in attracting adult rice pests, thereby providing a more scientific reference for the selection of monitoring devices for major rice pests. |
||||
|
Comments 2: [In my opinion, the manuscript needs to be revised by eliminating the introductory and overly general sections.] |
||||
|
Response 2: Thank you for pointing this out. We agree with this comment. We have made corresponding modifications, deleted overly general content, and cited a large number of internationally published literature related to this study as an explanation of the importance of conducting this study and as evidence of its scientificity and feasibility, making the conduct of this experiment and the results obtained appear more scientific, reasonable, and more readable (in green). |
||||
|
Comments 3: [It is also necessary to avoid repetitions between the written part and the illustrations.] |
||||
|
Response 3: Thank you for pointing this out. We strongly agree with this view. We have optimized the written part of the experimental results, focusing on the theme of this study and avoiding duplication between the written part and the illustrations. For some results expressed in the illustrations (which appear less important), we have deleted or simplified them in the written part. |
||||
|
Comments 4: [Focus attention on the objectives of the activities in favor of simple, field-applicable, cost-effective and effective indications.] |
||||
|
Response 4: Thank you for pointing this out. We strongly agree with this point of view and have reorganized the language in the written expression section to clarify the experimental results that this study most wants to present. |
||||
|
Comments 5: [In the attached PDF I tried to highlight some parts that should be shortened (in green) or that are the real scientific content (in red) that require a more in-depth reflection] |
||||
|
Response 5: Thank you for pointing this out. We have revised the article according to your highlighted points (please read the newly submitted manuscript). â‘ We have shortened and revised the Introduction (in green). â‘¡We cited more publicly published literature to demonstrate the results of this study and confirmed the accuracy and scientific nature of the conclusions of this study.
|
||||

Reviewer 2 Report
Comments and Suggestions for Authors
Review report
The manuscript presents a comparative field study on multi-source trapping devices combining color plates, pheromones, UV light, and cover modifications, aimed at monitoring major rice pests. The work is of practical importance for integrated pest management (IPM), as it addresses the efficiency of multi-source devices for simultaneous pest trapping while evaluating ecological safety. The findings on the contrasting responses of Cnaphalocrocis medinalis versus other pests are particularly valuable for device optimization. Overall, the manuscript is relevant, well-structured, and provides novel insights. However, some sections require clarification, deeper analysis, and improvements in presentation to enhance the manuscript’s scientific rigor and practical applicability.
- The abstract shows a very high level of similarity (≈99%) with previously published or submitted manuscripts on related topics. This raises concerns about originality and potential self-plagiarism. I strongly recommend that the authors thoroughly revise the abstract to present the study’s objectives, methods, key results, and conclusions in their own words, avoiding overlap in phrasing and structure. The abstract should be concise, clear, and sufficiently distinct from prior work to highlight the novelty and unique contribution of this study.
- The manuscript currently lacks a properly formulated title and introduction section.
- The description of the devices is overly technical and wordy, making it hard to follow. Abbreviations (cp, sp, sc, lc, ul, st, etc.) are introduced in bulk and reused excessively, which reduces readability.
- Although it mentions “three representative rice fields” at each site, it is unclear whether these represent true replicates or just observational placements. More detail on replication (number of traps per treatment, number of independent blocks) is needed.
- Distances between devices (8 m) are given, but justification is missing — is this enough to avoid cross-attraction or interference?
- No mention of pesticide use or agronomic practices in test fields, which could confound results.
- Some devices are tested only against specific pests ( suppressalis or C. medinalis), while others are tested against mixed pest complexes. This makes comparisons uneven and less systematic.
- Insects were surveyed weekly, but details are missing: How were insects identified (morphological keys? by whom?) Were all life stages counted or only adults? How were natural enemies classified?
- Only t-tests and one-way ANOVA are used, but the design is a multi-factorial field experiment (device type × site × season). More appropriate models (e.g., GLM, mixed-effects models) could handle block/site effects better.
- It is unclear whether sample size was sufficient for robust statistical power.
- The description of data transformation is vague (“logarithm or square root transformation”) without specifying which variables required transformation.
Author Response
|
Reply to reviewer 2 |
||
|
1. Summary |
|
|
|
Dear reviewer, thank you very much for taking the time to review this manuscript and for your valuable feedback. Your suggestions have been very helpful in revising and improving the manuscript. We have carefully considered and revised your valuable comments. Please read our detailed response below. |
||
|
2. Questions for General Evaluation |
Reviewer’s Evaluation |
Response and Revisions |
|
Does the introduction provide sufficient background and include all relevant references? |
Yes/Can be improved/Must be improved/Not applicable |
We cited more literature related to this study to increase the feasibility, necessity and scientificity of this experimental research topic (in green). |
|
Is the research design appropriate? |
Yes/Can be improved/Must be improved/Not applicable |
We accept your point of view. However, we do not believe there are major issues with the study design, but we do need more detailed elaboration and writing of the Experimental Design and Methods section.. |
|
Are the methods adequately described? |
Yes/Can be improved/Must be improved/Not applicable |
We agree with this comment. In the experimental methods section, we provide a detailed description and explain more details of the experimental methods involved in this study (in red). |
|
Are the results clearly presented? |
Yes/Can be improved/Must be improved/Not applicable |
|
|
Are the conclusions supported by the results? |
Yes/Can be improved/Must be improved/Not applicable |
|
|
Are all figures and tables clear and well-presented? |
Yes/Can be improved/Must be improved/Not applicable |
|
|
3. Point-by-point response to Comments and Suggestions for Authors |
||
|
Comments 1: [The abstract shows a very high level of similarity (≈99%) with previously published or submitted manuscripts on related topics. This raises concerns about originality and potential self-plagiarism. I strongly recommend that the authors thoroughly revise the abstract to present the study’s objectives, methods, key results, and conclusions in their own words, avoiding overlap in phrasing and structure. The abstract should be concise, clear, and sufficiently distinct from prior work to highlight the novelty and unique contribution of this study.] |
||
|
Response 1: Thank you for pointing this out. â‘ The main content is our original work and is not plagiarized or copied from previously published literature.The high similarity (≈99%) you claim is due to the fact that this research paper was previously published as a preprint (DOI: https://doi.org/10.21203/rs.3.rs-7193580/v1). We previously submitted our manuscript to the Journal of Pest Science and published a preprint, but unfortunately, it was rejected. â‘¡However, in response to your subsequent questions and suggestions for revision, we have adjusted the content of the abstract; following the journal editor's instructions, we have added a brief summary before the abstract. |
||
|
Comments 2: [The manuscript currently lacks a properly formulated title and introduction section.] |
||
|
Response 2: Thank you very much for pointing this out. We agree with this view. â‘ Based on your and another reviewer's suggestions, we have revised the written content of this article to use clearer and simpler language. We will use " Synergistic attraction and ecological effects of multi-source physical and chemical trapping methods with different mechanism combinations on rice pests" as the title of this article. This is a decision we made after careful consideration. We have indeed given careful consideration to the title. â‘¡Regarding the introduction, you and another reviewer agree. Thank you very much. We have revised it accordingly, deleting some overly general references and citing a large number of relevant references to this study to illustrate the importance of conducting this study and to support its scientific and feasibility. This makes the conduct of this experiment and the results obtained more scientific, reasonable, and readable (in green). |
||
|
Comments 3: [The description of the devices is overly technical and wordy, making it hard to follow. Abbreviations (cp, sp, sc, lc, ul, st, etc.) are introduced in bulk and reused excessively, which reduces readability.] |
||
|
Response 3: Thank you for pointing this out. â‘ In fact, when we named the device combinations (Light-shielding/Light-transmitting, color plates, insect sex pheromones, transmitting(shielding) light covers, and solar-powered automatic insect-attracting ultraviolet lamps), we also considered that the descriptions were too technical and lengthy to be understood by readers. â‘¡In addition, we have detailed each experimental trapping device combination in Section(2. Materials and Methods), using Figure 1 and Table 1 to clearly illustrate the concepts and provide a more intuitive and clear understanding. These two approaches are intended to enhance readability and facilitate understanding (in red). Therefore, after considerable consideration, we decided to use simple abbreviations (cp, sp, sc, lc, ul, st) to more clearly illustrate the trapping effects of different trapping device combinations. |
||
|
Comments 4: [Although it mentions “three representative rice fields” at each site, it is unclear whether these represent true replicates or just observational placements. More detail on replication (number of traps per treatment, number of independent blocks) is needed.] |
||
|
Response 4: Thank you for pointing this out. The "three representative rice fields" were located in different rice-growing areas at the same location, but not adjacent to each other, theoretically representing three true experimental replications. Within each rice field, we randomly set up different treatments (different combinations of traps) in a randomized arrangement. The figure with n=3 represents the results of three experimental replications. |
||
|
Comments 5: [Distances between devices (8 m) are given, but justification is missing — is this enough to avoid cross-attraction or interference?] |
||
|
Response 5: Thank you for pointing this out. The minimum distance between devices (8 meters) was determined based on the illumination and attracting range (diameter) of the 0.36W solar-powered automatic insect-attracting ultraviolet lamps (ul) and insect sex pheromones used in the field trial at night. In practice, the minimum distance between devices (8 meters) was sufficient to avoid cross-contamination between treatments. |
||
|
Comments 6: [No mention of pesticide use or agronomic practices in test fields, which could confound results.] |
||
|
Response 6: Thank you for pointing this out. We greatly appreciate your comment and appreciate your kind suggestion and timely reminder. We have added relevant information on these aspects to the Section(2. Materials and Methods) of the paper. The complete experiment in this study was conducted from start to finish without the use of pesticides or agricultural practices (in blue). |
||
|
Comments 7: [Some devices are tested only against specific pests (C. suppressalis or C. medinalis), while others are tested against mixed pest complexes. This makes comparisons uneven and less systematic.] |
||
|
Response 7: Thank you for pointing this out. Here is our explanation: ①As shown the Section(2. Materials and Methods,Table 1), this study focused on the fact that different rice growth stages produce different major pest species, and therefore different statistical targets. Based on the occurrence and prevalence patterns of major rice pests at different growth stages and throughout the year, we conducted experimental research using two insect pheromone lures (C. suppressalis and C. medinalis, sp(Cs) and sp(Cm)) along with different optimized combinations (Figure 1 and Table 1). Our goal was to develop a trapping device more suitable for monitoring major rice pests throughout their entire growth period. ②C. suppressalis and C. medinalis are not specific pests, but rather the primary pests monitored in Test number 1 (see the first row of Table 1 for explanations, in red). The results of Experiment 1 were a key inspiration for this study and the source of the underlying ideas. Based on the field population, other pests were less important in Experiment 1, but they are the subject of future research. Therefore, the comparison between the final results of the trials is balanced and systematic. |
||
|
Comments 8: [Insects were surveyed weekly, but details are missing: How were insects identified (morphological keys? by whom?) Were all life stages counted or only adults? How were natural enemies classified?] |
||
|
Response 8: Thank you for pointing this out. Here is our explanation: â‘ The experimenters regularly go to the fields every week to conduct on-site surveys and statistics on each device, and identify, classify and count the pests on the sticky insect boards according to the typical morphological characteristics of different rice pests, and replace new sticky insect boards. â‘¡The trapping device is located above the rice canopy, and the rice pests trapped are adults, so we only counted the adults (within one growth period of rice). â‘¢The classification method of natural enemies is as follows: based on the typical morphological characteristics of natural enemies or beneficial insects of different rice pests, the experimenters identify the types of natural enemies or beneficial insects on the sticky insect boards in the field (species), and finally classify and count each natural enemy type as belonging to a certain "family or genus". In Section2 (Materials and Methods), we have written this down in more detail (in red). |
||
|
Comments 9: [Only t-tests and one-way ANOVA are used, but the design is a multi-factorial field experiment (device type × site × season). More appropriate models (e.g., GLM, mixed-effects models) could handle block/site effects better.] |
||
|
Response 9: Thank you for pointing this out. Here is our explanation: â‘ As shown in the second section(Materials and Methods) of this article, Figure 1, and Table 1, the types and severity of major rice pests vary depending on the rice growth stage and regional variations. Based on the types and regional characteristics of major rice pests, we conducted two consecutive experiments at two sites using different trapping combinations tailored to the rice growth stage and major pest species. Each independent experimental design (Test numbers 1, 2, 3, 4, 5, and 6) was based on a single-factor experimental design (in red) from Test Site 1 or Test Site 2 during the same season. In this experiment, we only analyzed and compared the differences in the number of major rice pests captured by different trapping combinations based on the characteristics of the major rice pests. Therefore, only Only t-tests and one-way ANOVA analysis of variance were required. â‘¡As shown in Table 1, except for some individual treatments in (Test site 1 of Number 3), such as the Light-transmitting light green sticky plate device ( cp+lc ), which did not use a sex attractant core, sp(Cs) or sp(Cm) was used in all other cases, but they were all independent experiments(Test numbers 1, 2, 3, 4, 5, and 6). |
||
|
Comments 10: [It is unclear whether sample size was sufficient for robust statistical power.] |
||
|
Response 10: Thank you for pointing this out. Here is our explanation: ①The experiment was conducted in an open field, which is a dedicated intensive rice cultivation area (the previous crop was winter rapeseed) and also an area where major rice pests often occur. ②All insects trapped in the experiment were adults. In the rice-growing region where the experiment was conducted, major pests such as the Chilo suppressalis occur annually, resulting in a large insect population base that meets the experimental requirements. The pest sample size is sufficient to obtain robust statistical results. ③Migratory pests such as rice planthoppers and rice leaf rollers are affected by factors such as the timing and number of arrivals, and the rice growing period after arrival. Their occurrence fluctuates significantly from year to year, leading to fluctuations in sample sizes. In this study (2023 and 2024), the population sizes of the main pests met the experimental requirements and were sufficient to obtain robust statistical results. We will make revisions to the manuscript and include statistical analysis values ​​in the 3. Results. We have already explained the above contents in section 2.2 Test Location and Basic Information (in red). |
||
|
Comments 11: [The description of data transformation is vague (“logarithm or square root transformation”) without specifying which variables required transformation.] |
||
|
Response 11: Thank you for pointing this out. Here is our explanation: The manuscript already explains the specific methods for significance analysis, and while logarithmic or square root transformation of the data is mentioned, the data used in this study are actual field data and were not logarithmically or square root transformed. The trapping yields shown in the figure are the average of three replicates of the field experiment. Some traps did indeed catch less than one insect on average. In 2.4. Data Analysis Methods, we performed deletions and additions (in yellow). |
||

Round 2
Reviewer 1 Report
Comments and Suggestions for Authors
The Authors of the manuscript have revised the previous version. The changes made have improved the work, although not all the suggestions I made in the first round of revisions were always accepted. Overall, the current version of the work is quite interesting, especially from a technical standpoint; the contribution of new scientific knowledge is less important.
Reviewer 2 Report
Comments and Suggestions for Authors
The authors have addressed all of my comments thoroughly. I have no further remarks. I recommend the manuscript for publication and extend my congratulations to the authors.